# Facile and Scalable Synthesis and Self-Assembly of Chitosan Tartaric Sodium

**DOI:** 10.3390/polym14010069

**Published:** 2021-12-25

**Authors:** Sixuan Wei, Rujie Peng, Shilong Bian, Wei Han, Biao Xiao, Xianghong Peng

**Affiliations:** Key Laboratory of Optoelectronic Chemical Materials and Devices (Ministry of Education), School of Optoelectronic Materials & Technology, Jianghan University, Wuhan 430056, China; WSX15927474220@163.com (S.W.); 18007161071@163.com (R.P.); bianshilong5200@163.com (S.B.); e781206379@gmail.com (W.H.)

**Keywords:** chitosan, tartaric acid, nanocrystal, self-assembly, evaporation, dehydration of crystallization

## Abstract

Chitosan-based nanostructures have been widely applied in biomineralization and biosensors owing to its polycationic properties. The creation of chitosan nanostructures with controllable morphology is highly desirable, but has met with limited success yet. Here, we report that nanostructured chitosan tartaric sodium (CS-TA-Na) is simply synthesized in large amounts from chitosan tartaric ester (CS-TA) hydrolyzed by NaOH solution, while the CS-TA is obtained by dehydration-caused crystallization. The structures and self-assembly properties of CS-TA-Na are carefully characterized by Fourier-transform infrared spectroscopy (FTIR), nuclear magnetic resonance spectroscopy (^1^H-NMR), X-ray diffraction (XRD), differential scanning calorimeter (DSC), transmission electron microscopy (TEM), a scanning electron microscope (SEM) and a polarizing optical microscope (POM). As a result, the acquired nanostructured CS-TA-Na, which is dispersed in an aqueous solution 20–50 nm in length and 10–15 nm in width, shows both the features of carboxyl and amino functional groups. Moreover, morphology regulation of the CS-TA-Na nanostructures can be easily achieved by adjusting the solvent evaporation temperature. When the evaporation temperature is increased from 4 °C to 60 °C, CS-TA-Na nanorods and nanosheets are obtained on the substrates, respectively. As far as we know, this is the first report on using a simple solvent evaporation method to prepare CS-TA-Na nanocrystals with controllable morphologies.

## 1. Introduction

Chitosan, a cationic polysaccharide composed of β-(1–4) linked 2-acetamido-2-deoxy-β-d-glucopyranose and 2-amino-2-deoxy-β-d-glycopyranose, is an alkaline deacetylation product of chitin, the second most abundant polysaccharide, which mainly comes from the exoskeletons of crustaceans, insects, beetles, as well as the cell walls of fungi [1]. Many of the applications of chitosan in several fields are based on its biological and excellent cationic properties [2,3], including biocompatibility [4], low immunogenicity, low or no toxicity, and antibacterial and moisture retentive properties [5,6]. Chitosan-based nanomaterials (such as nanogels, nanofibers, and nanocrystals) have been paid increasing attention due to their size-specific and free amine properties. In previous studies, chitosan nanogels were usually prepared (1) by non-covalent cross-linking with sodium tripolyphosphate, (2) by chemical cross-linking with glutaraldehyde, genipin and dicarboxylic acid, (3) by electrostatic interactions through changing the pH of the medium [7]. As demonstrated in a recent study, chitosan tartaric acid based-nanogels can be also prepared in a reverse microemulsion system through a condensation reaction between carboxylic groups of dicarboxylic acids and amino groups of chitosan, in which 1-ethyl-3-(3-dimethylaminopropyl)-carbodiimide (EDC) and N-hydroxysuccinimide (NHS) were used as coupling agents [8,9]. Although, they could effectively deliver vitamin B_12_ and blue dextran, the obtained nanogels can only be dispersed in an acid solution, limiting their scope of application. According to the research, chitosan nanostructures (e.g., nanowires, nanotubes, and nanorods) could be obtained during the electrochemical synthesis, through tuning the reaction conditions to adjust the hydrogen-bonding interactions of chitosan. The chitosan was degraded by ultrasound in an acidified propylene carbonate solution to obtain chitosan embryo, then the chitosan nanostructures were produced under an electric field [10]. Although this synthetic process was simple and could be performed under mild and usual experimental laboratory conditions, the resulting chitosan nanostructures could only exist on the surface of the electrode. Other chitosan nanofibers were prepared using electrospinning method under high-voltage conditions (15–25 kV); nevertheless, they were all amorphous with only thread-like morphology. Other recent studies indicated that chitin nanostructures with amine groups were excellent scaffolds for bone regeneration and cell adhesion, and in particular, chitosan with acetamide groups was the main subject [11,12]. Currently, carboxylated chitosan has been prepared from carboxylated chitin by alkaline deacetylation. The carboxylated chitin was produced with ammonium persulfate as a mild oxidant [13]. Due to the high crystallinity of natural chitin structure, this carboxylated chitosan showed nanorod morphology in the aqueous solution. Furthermore, studies showed that the nanostructures of chitin and chitosan were mostly dispersed in water, and the accumulation of nanoparticles could affect the cationic properties of chitosan, limiting its further applications in the solid state. Meanwhile, the structured chitosan-based nanomaterial pattern will be beneficial to the applications of nanoscale carriers and sensors.

In nature, based on the regular nanostructures as building blocks, many remarkable biological materials are formed by self-assembling the repetitive discrete components into higher-order structures [14], such as squid pens and crab shells [15], butterfly wings [16,17], and pearls [18]. Notably, the biomaterials mentioned above have excellent mechanical properties, anisotropy and even structural colors due to their non-covalently linked hierarchical architectures [14]. In particular, photonic hydrogels could be prepared by using twisted mesoporous chitosan nanofibrils as a precursor for acetylation and a platform for templating poly(methylmethacrylate) [19]. As reported in previous studies, chitosan-based photonic crystals with a layered structure and color tunability could be also fabricated using surface binding and polymerization method. During the synthesis, polymethacrylic acid was deposited on a morpho butterfly wing template [16]. To date, inspired by nature, the specifically designed photonic crystals based on chitosan nanofibrils can respond to organic solvent or moisture [20,21,22]. Similar to chitosan-based nanostructures, self-assembled peptide nanostructures possess unique physical and biological properties and have broad application prospects in the field of electronic devices and functional molecular recognition [23]. Polypeptides with unique helical structures have been self-assembled due to the strong hydrogen bonds between amino and carboxyl groups in polypeptide molecules and other non-chemical bonds [24]. To date, controllable morphologies and patterns of polypeptides have been realized by evaporating dehumidification solution on template substrates [23]. Based on this method, we intended to synthesize carboxylated derivatives of chitosan while retaining their semi-crystalline structure, and to investigate their controllable morphologies.

Owing to their special properties, the creation of chitosan nanostructures with controllable morphology is highly desirable, but has had limited success so far. Therefore, we intended to synthesize chitosan tartaric sodium (CS-TA-Na) which has –NH_2_ and –COO– groups and a semi-crystalline structure by a simple synthesis method. The controllable morphologies of CS-TA-Na nanostructures were prepared using a solvent evaporation method. To the best of our knowledge, this is the first report using a facile and scalable method to prepare a mass of CS-TA-Na nanostructures. The study on chitosan-based nanostructures will promote the development of biomineralization and biosensors.

## 2. Experimental Section

### 2.1. Materials

Chitosan (95% deacetylation, the viscosity of 1 wt% chitosan solution was 100–200 mpa·s and the source was shrimp shell as reported by the supplier) was purchased from Macklin Reagent Biochemical Co. Ltd., Shanghai, China. The other agents were purchased from Shanghai Sinopharm Reagent Co., Shanghai, China. All reagents were analytical grade and used without further purification.

### 2.2. Sample Preparation

The synthetic method was prepared according to previous work [8,9] with some modifications. 0.5 wt% chitosan tartaric acid aqueous solution was obtained by dissolving the chitosan and 7.5 wt% tartaric acid in water, then the solution was poured into an open container and placed in the 90 °C air drying for 12–24 h to realize dehydration caused crystallization. After that, the sample was washed with ethanol by 6.5 wt% concentration of the dispersion to obtain the powder, named chitosan tartaric ester (CS-TA). We mixed 2 wt% NaOH aqueous solution and the ethanol dispersion of CS-TA at same volumes accompanied with stirring, then the solution was centrifuged and the centrifugal precipitate was washed with ethanol, then dried to obtain the CS-TA-Na powder.

Glass and polytetrafluoroethylene (PTFE) were washed with deionized water and ethanol, and then soaked in anhydrous ethanol for later use. CS-TA-Na powder was dispersed in pH = 7–8 solution to obtain the CS-TA-Na dispersed aqueous solution, which was then drop-cast on the substrate surface. The CS-TA-Na substrate was dried at different temperatures, and soaked in anhydrous ethanol for 30 min, and then dried for observation.

### 2.3. Instrumentation

The ATR-FT-IR spectra of the samples were recorded on FT-IR spectrometer (TENSOR 27, BRUKER, Billerica, MA, USA) with KBr powder as background, resolution set as 4 cm^−^^1^, average number of scans set as 16 and ranging from 4000 cm^−1^ to 600 cm^−1^. ^1^H-NMR spectra for the samples in a mixed solvent of CF_3_COOD and D_2_O (2:98 /*w*_1_:*w*_2_) were acquired on NMR spectrometer ( AVANCE NEO 400M and AVANCE NEO 500M, BRUKER, Billerica, MA, USA). X-ray diffraction (XRD) patterns were obtained using an XRD diffractometer (D8-Advance, BRUKER, Billerica, MA, USA) with Cu Kα radiation (λ = 0.15406 nm). The sample morphologies were characterized by transmission electron microscopy operated at 200 kV (HT7700 EXALENS, HITACHI, Tokyo, Japan), scanning electron microscope (SU8010, HITACHI, Tokyo, Japan)with sample coated with gold, and polarizing microscope (BX41-LED, OLYMPUS, Tokyo, Japan).

## 3. Results and Discussion

It is well known that functional groups are the reason why chitosan has many attractive properties. Therefore, the ATR-FTIR test was used to explore the changes of functional group during the synthesis of CS-TA and CS-TA-Na. Figure 1 shows the FTIR spectra of chitosan, CS-TA and CS-TA-Na. The spectrum of pure chitosan exhibited characteristic peaks at 1650 cm^−^^1^ and 1599 cm^−^^1^ originating from amide I (C=O stretching) and amide II (–NH_2_ stretching) of N-acetylglucosamine and N-glucosamine units, respectively (Figure 1A). Compared to chitosan, CS-TA spectrum presented a new peak at 1729 cm^−1^, which is attributed to the C=O stretching band [8,25]. It has been confirmed in previous studies that C6 alcohols and the –NH_2_ groups of chitosan react with the carboxylic groups of tartaric acid during the dehydration caused crystallization (Figure 1B) [8,26]. What is noteworthy is that the CS-TA-Na spectrum demonstrated a strong absorption peak at 1600 cm^−1^ derived from the symmetric and asymmetric –COONa and –NH_2_ bonds, indicating that CS-TA was hydrolyzed in NaOH aqueous solution to form salt bonds with –COONa groups (Figure 1C) [27,28]. In short, the ATR-FTIR results revealed that CS-TA-Na was featured in both carboxyl and amino functional groups.

In order to accurately determine the chemical structure of the reactants and products, ^1^H-NMR measurement was carried out. Figure 2 demonstrates typical ^1^H-NMR spectra of chitosan and CS-TA-Na in CD_2_COOD/D_2_O = 2:98 (*w*_1_:*w*_2_) solvent. The ^1^H-NMR spectra of CS-TA-Na exhibited an isolated peak at 4.15–4.21 ppm and 1.0–1.1 ppm, corresponding to the resonance of α-H protons of CS-TA-Na sample (Figure 2B). The presence of this resonance indicates the successful reaction between chitosan and tartaric acid. Meanwhile, the peak at 2.65 ppm was assigned to the three protons of N-acetylglucosamine (GlcNAc) units and −CH_2_COONa of glucosamine (GlcN) [26], while the peak at 3.1–3.5 ppm corresponds to H-2 proton of GlcN of chitosan and CS-TA-Na. The substitution degree (DA) of chitosan amino acid substituted by tartaric acid was calculated using H area and –NH_2_ area on the pyranose ring, and the resulting DA value was 10.32%. The DA was calculated based on the ^1^H-NMR spectrum considering the peak areas of the α-H of tartaric acid (4.15–4.2 ppm) and the GlcNAc (3.15–3.5 ppm) using the following equation [29]:DA (%) = [I_α-H_/(I_GlcNAc_/5)] × 100(1)

I_α-H_ and I_GlcNAc_ are peak areas of the α-H of tartaric acid (4.15–4.2 ppm) and GlcNAc (3.15–3.5 ppm). The assignments and chemical shifts of the ^1^H-NMR signals are given as follows: chitosan ^1^H-NMN (D_2_O/CD_3_COOD, 500 MHz, 20 °C): d = 4.3–4.4 (1-H of GlcN), 3.1–3.4 (3-H, 4-H, 5-H, 6-H, 2-H of GlcNAc), 2.65 (2-H of GlcN), 1.65 (HN-COCH_3_). CS-TA-Na^1^H-NMN (D_2_O/CD_3_COOD): d = 4.4–4.5 (1-H of GlcN), 3.15–3.5 (3-H, 4-H, 5-H, 6-H of GlcNAc), 2.75 (2-H of GlcN), 1.65 (HN-COCH_3_), 4.15–4.2 (α-H of tartaric acid), 2.9–3.11 (α′-H of tartaric acid) [28,29].

The XRD test can further obtain the crystallization information of the materials. XRD patterns of chitosan, CS-TA and CS-TA-Na are shown in Figure 3. Compared to that of chitosan, the intensities of the crystalline diffraction peak at 2*θ* = 19.9° for 110 reflection increased, indicating that there were relatively regular lattices in CS-TA and CS-TA-Na (Figure 3A,B). Interestingly, a new peak at 2*θ* = 37.4° was observed in the pattern of CS-TA-Na, which is attributed to the stronger interaction between Na^+^ and NH⋯O=C groups (Figure 3B) [30]. As previously reported, the nanoparticles of chitosan crosslinked with tartaric acid were essentially amorphous, because the crystal domain of chitosan was disturbed by the carboxyethyl groups [26]. In this work, the easy crystallization properties of tartaric acid would cause CS-TA to form relatively regular lattices.

Figure 4 shows the schematic of the synthesis process of CS-TA-Na. Firstly, chitosan was dissolved in tartaric acid solution, and its amine group protonated in the acid aqueous solution. Chitosan tartrate acid solution is a clear solution and stable for several months. Secondly, the chitosan tartaric acid aqueous solution was cast on the substrate, and then, dehydrated to cause crystallization and form CS-TA. As far as we know, the tartaric acid would crystallize during the dehydration and evaporation process, the crosslinking reaction and entanglement effects between chitosan and tartaric acid would cause the CS-TA to form a tightly stacked and long-range crystal structure. Such a result was proved by XRD results. Thirdly, CS-TA was hydrolyzed by NaOH solution to form salt linkages in between CS-TA-Na when CS-TA was added to NaOH solution, in which CS-TA-Na processes –NH_2_ and –COONa groups. It is noteworthy that the regular crystal lattice of CS-TA-Na remained during the hydrolysis process at room temperature. To the best of our knowledge, this is the first report that the CS-TA-Na is synthesized using water evaporation and crystallization method without a complex purification process, and with this method, gram-grade CS-TA-Na can be obtained under the usual laboratory conditions.

Figure 5 presents the photos of CS-TA-Na powder, CS-TA-Na water dispersion, and the TEM image of CS-TA-Na nanocrystals. It is worth noting that CS-TA-Na demonstrated a high yield and favorable water dispersibility. Meanwhile, the TEM image shows that the CS-TA-Na were about 20–50 nm in length and 10–15 nm in width. Interestingly, due to the hydrogen bonding interactions within and in between CS-TA-Na molecules, CS-TA-Na was prone to form a multilayer structure of polycrystals, leading to the parallel arrangement of CS-TA-Na nanocrystals and nanorods [31]. Such parallel structures are beneficial to self-assembly, resulting in ordered CS-TA-Na structures.

Figure 6 demonstrates the morphologies of self-assembled CS-TA-Na on glass and PTFE substrate at 4 °C, 25 °C and 60 °C, respectively. As shown in Figure 6G, the morphologies presented a dependence on the water evaporation rate and the surface characteristics of the substrate. When the water evaporation was slow (at 4 °C), CS-TA-Na had ample time to adjust and self-assemble, resulting in a more ordered structure. The assembled nanorods were more than 1 μm in length and 50 nm in width (Figure 6A,D) [31]. When the air-drying temperature increased, those dispersed CS-TA-Na nanocrystals tended to assemble quickly and interlace. At 25 °C, interlaced nanorods and nanosheets were observed (Figure 6B,E). Further increasing the drying temperature to 60 °C could lead to the flower-like pattern composed of vertical nanocrystals and nanosheets, as shown in Figure 6C,F. Obviously, there existed the vertical CS-TA-Na assembly on both the glass and PTFE substrate at 60 °C, suggesting that the morphologies of the patterns were mostly affected by the strong vertical convection originating from the higher rate of solvent evaporation. Therefore, the three-dimensional orders of the CS-TA-Na patterns could be controlled by tuning the rate of water evaporation and the surficial conditions of the substrate [23,32].

The self-assembly process of CS-TA-Na follows the nucleation and growth mechanism, as shown in Figure 7. First, CS-TA-Na nanocrystals dispersion was dripped onto the substrate to form a uniform liquid layer (Figure 7A). Then, nanocrystals moved towards the edge to form the seeds for nucleus growth on the substrate edge during the solvent evaporation. Finally, nanocrystals continued to grow around the initial nuclei to form the nanocrystals’ domain and dendrimer. The bright birefringence pattern of the nanocrystals with the main dendritic morphology was observed after the dispersion droplets were cast and evaporated at 60 °C for 60 s (Figure 7B,C). Meanwhile, an obvious coffee ring was more proof that the self-assembly of CS-TA-Na nanocrystals was influenced by fluid mechanical effects [33].

## 4. Conclusions

In summary, we presented, for the first time, a facile and scalable method by which we can readily produce a mass of CS-TA by solvent-evaporation caused crystallization, in which tartaric acid was used as the crystallization and the crosslinking agent. Subsequently, CS-TA-Na featured in surface carboxyl, amine functionalities and regular crystal lattice structure was prepared by hydrolysis with NaOH aqueous solution. The resulting CS-TA-Na nanocrystals were 20–50 nm in length and 10–15 nm in width in solution. The nanostructure patterns of CS-TA-Na (including nanosheets, nanorods) were then obtained by dripping its aqueous dispersions on various substrates and subsequently evaporating the solvent. Meanwhile, the exact shape and morphology of the nanostructures can be controlled by the air-drying temperature. As a result, the favorite CS-TA-Na nanosheets standing on the substrate were acquired at 60 °C on various substrates. It is the first time that self-assembly nanostructures with controllable morphology have been reported for chitosan. Importantly, this work proves a simple route to prepare chitosan-based nanostructure patterns.

## Figures and Tables

**Figure 1 polymers-14-00069-f001:**
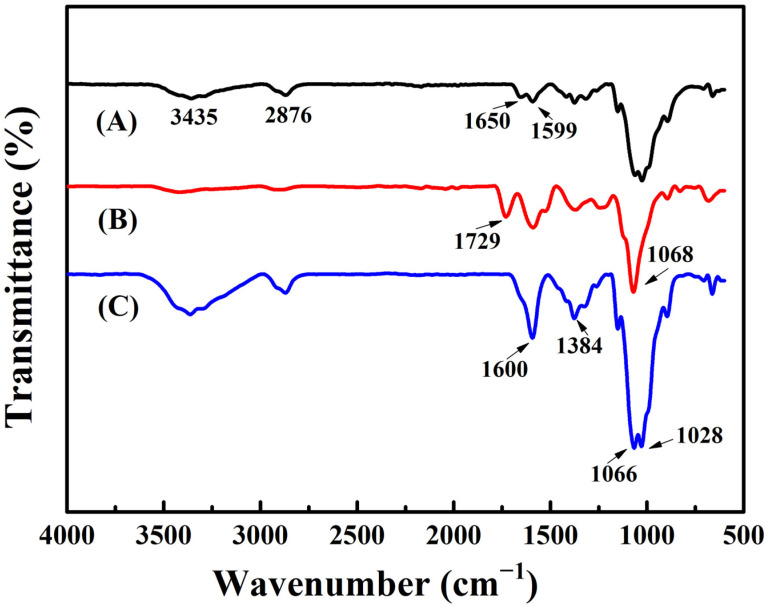
Attenuated total reflectance Fourier transform infrared (ATR-FTIR) spectra of (**A**) chitosan (CS), (**B**) chitosan tartaric ester (CS-TA) and (**C**) chitosan tartaric sodium (CS-TA-Na).

**Figure 2 polymers-14-00069-f002:**
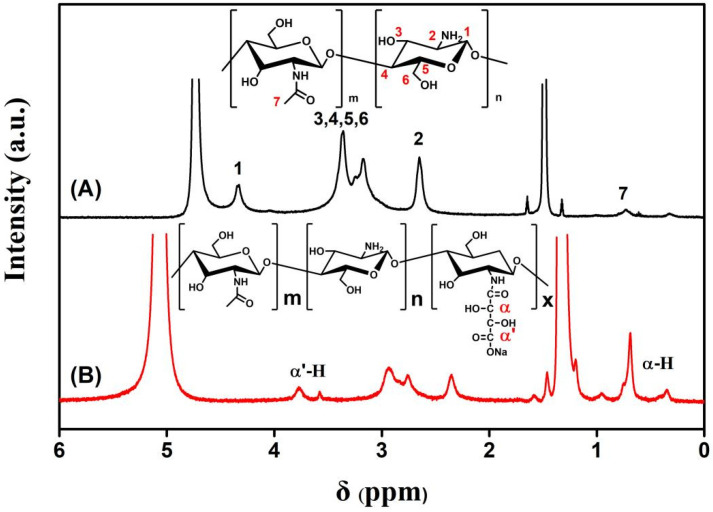
^1^H-NMR (nuclear magnetic resonance) spectra of (**A**) chitosan and (**B**) CS-TA-Na in CD_2_COOD/D_2_O = 2:98 (*w*_1:_*w*_2_) solvent.

**Figure 3 polymers-14-00069-f003:**
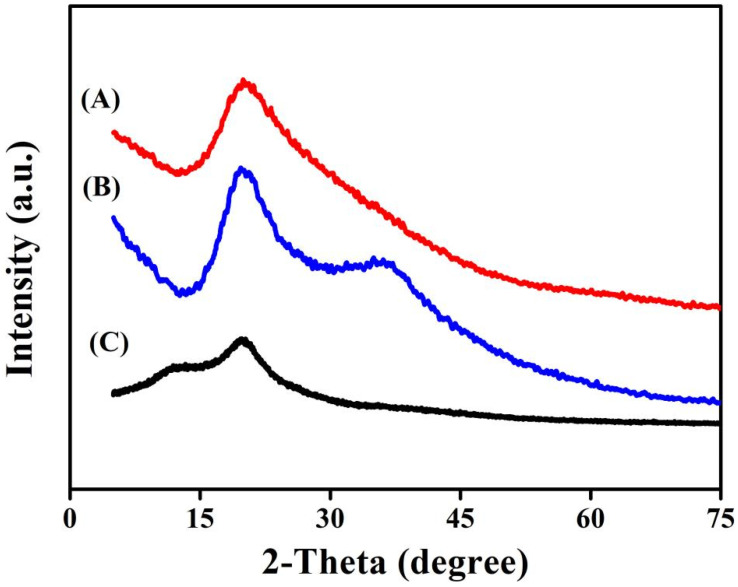
X-ray diffraction (XRD) patterns of (**A**) CS-TA, (**B**) CS-TA-Na, (**C**) chitosan.

**Figure 4 polymers-14-00069-f004:**
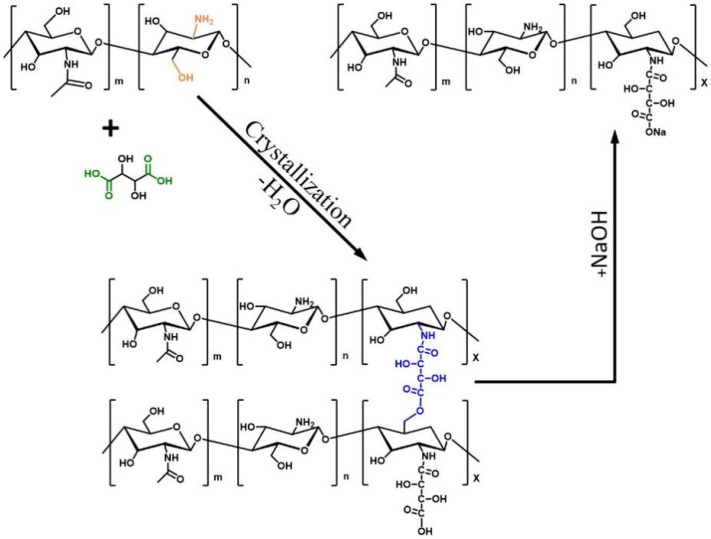
Schematic of the synthesis process of CS-TA-Na.

**Figure 5 polymers-14-00069-f005:**
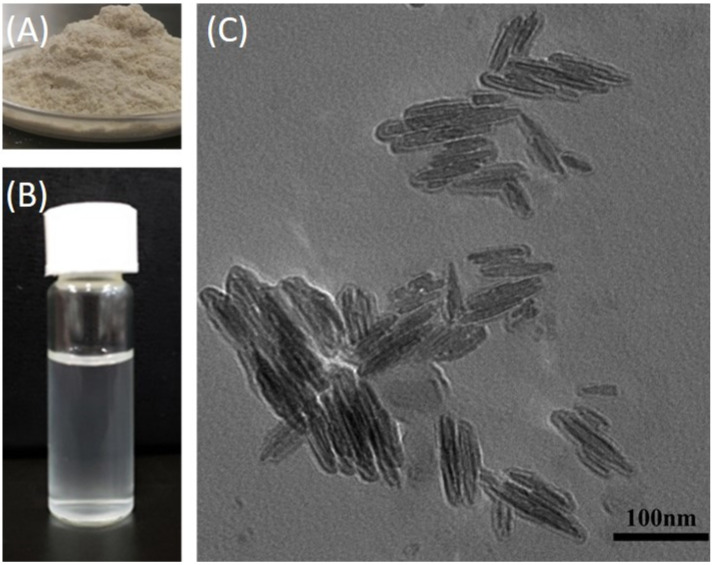
Photo of (**A**) CS-TA-Na powder and (**B**) its aqueous solution. (**C**) Transmission electron microscope (TEM) image of CS-TA-Na nanocrystals.

**Figure 6 polymers-14-00069-f006:**
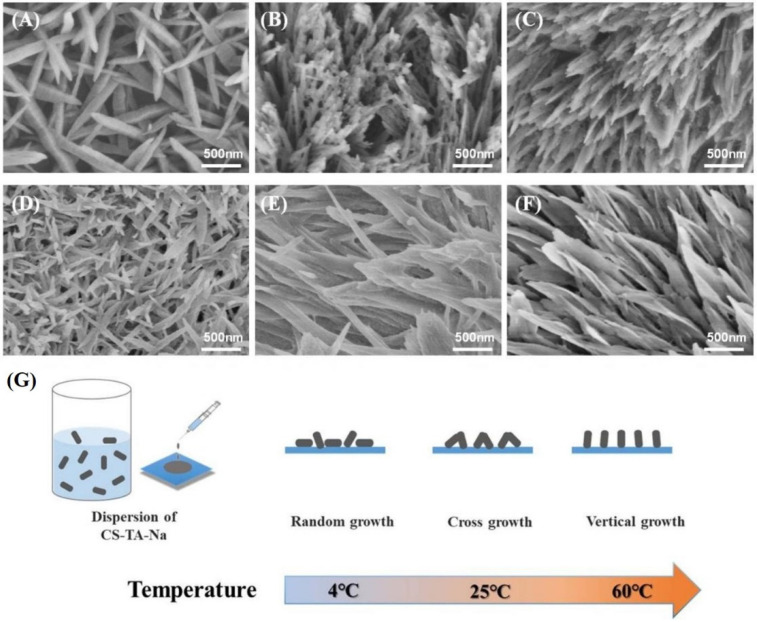
Scanning electron microscope (SEM) images: 5 μL 0.05 wt% CS-TA-Na dispersion drop-cast on glass substrate at (**A**) 4 °C, (**B**) 25 °C and (**C**) 60 °C, on PTFE substrate at (**D**) 4 °C, (**E**) 25 °C and (**F**) 60 °C. (**G**) Schematic of self-assembly of CS-TA-Na at different temperatures.

**Figure 7 polymers-14-00069-f007:**
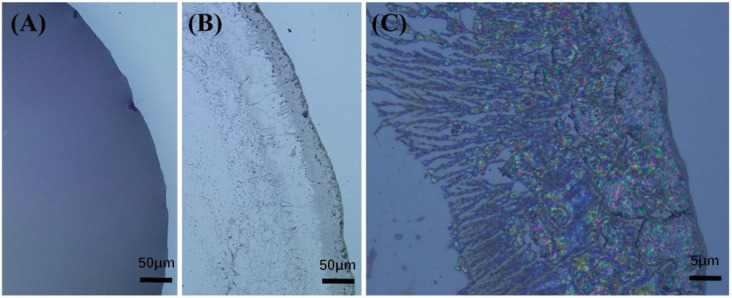
Polarizing optical microscope (POM) images of 0.05 wt% CS-TA-Na water dispersion self-assembled at 60 °C with the lapse of time: (**A**) 0 s, (**B**) 60 s, (**C**) enlarged view of (**B**).

## Data Availability

The data that support the findings of this study are available from the corresponding author, X.P., upon reasonable request.

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
