# Peer review of "Facile and Scalable Synthesis and Self-Assembly of Chitosan Tartaric Sodium"

_polymers, 2021, doi:10.3390/polym14010069_

Round 1

Reviewer 1 Report

The manuscript presents the development of a new and rapid way to synthesize chitosan-tartaric sodium nanostructures (nanorods) through the chitosan-tartrate ester chemical hydrolysis and air-drying crystallization procedure. This method is standard in polypeptide self-assembly nanostructure synthesis but is the first time that has been reported for chitosan. I found a high degree of novelty with well-supported results from the experiments. However, several aspects need attention before the publication. In the first place, the abstract needs a more thorough background and explanation of why it is so important to find new ways to synthesize chitosan-nanostructures. Include relevant results and conclusions. Avoid a list of characterization techniques.

In the introduction, it is recommendable to add all the references to each example or case. Do not group the references. Check the style for the references if they follow the journal guidelines accordingly. Close with the aim of the work and the contribution to the knowledge field. The materials and methods section needs to present all the reagents and equipment (manufacturer, city, country). Please, cite the original methodology source for the chitosan functionalization. The molecular weight of the chitosan used as well as the source needs to be presented. Please, clarify the DA methodology.

Not all the readers are experts and need a more elaborated explanation for the DA calculation (including the equation used). There are several typos across the manuscript that need attention. For example, there is a point after the word Figure 1 (that should not be bold). Delete that. Check that across the manuscript. Line 209: should be PTFE. In the conclusions, why do you believe that the present results are scalable? There are several things to be considered before that statement becomes a reality. It is not clear how did you get the size of the nanocrystals in the results. Finally, what is the main contribution of the work to the knowledge field? Highlight them in the conclusions.

Reviewer 2 Report

Manuscript polymers-1483228 refers to the preparation of a sodium salt of chitosan crosslinked with tartaric acid. The manuscript is flawed by over-interpretation of the results and lack of essential pieces of information. The manuscript needs to be carefully re-written before being re-considered for publication.

General comments.

Authors claimed an ester formation and named the chitosan crosslinked with the tartaric acid chitosan tartaric ester (CS-TA). However, the authors did not present enough evidence related to ester formation. The FTIR spectra present a shoulder at 1730 cm-1, which could be related to ester bond formation. However, in reference 8 (Pujana et al., water-dispersible pH-responsive chitosan nanogels modified with biocompatible crosslinking agents. Polymer 2012, 53, 3107-3116), cited by the authors, the shoulder from 1729 cm-1 is related to the ester formation ”between chitosan and unreacted N-hydroxysuccinimide units.” The authors from reference 8 used N-hydroxysuccinimide for tartaric acid activation.

There are missing several essential pieces of information. For example: source of the chitosan (marine, fungal, insect?), the molecular mass of the chitosan, degree of (de)acetylation of chitosan, the pattern of chitosan de-acetylation. Authors have access to RMN; therefore, I am wondering why they did not make such analyses? For example, the pattern of chitosan de-acetylation could help their hypothesis regarding ester formation – the length of the tartaric acid molecules is not enough to react with two adjacent -NH2 groups. Another example, molar ratio between chitosan and tartaric acid is not calculated. The concentration of the dispersion of the claimed CS-TA in ethanol is not presented. How much 2% NaOH was dropped in the dispersion of the CS-TA in ethanol?  

The information from cited article is not properly present. For example, in L171-L172, it is mentioned ”As previously reported, the nanoparticles of chitosan crosslinked with tartaric acid were essentially amorphous, because the crystal domain of chitosan was disturbed by the carboxyethyl groups” and is cited reference 25 -   Qin et al. Self-assembly of stable nanoscale platelets from designed elastin-like peptide – collagen-like peptide bioconjugates. Biomacromolecules 2019, 20, 1514-1521. This reference 25 is not related to chitosan. I suppose a mistake in numbering (that is not compatible with the rigorous scientific approach). I checked reference 26. Huang et al. Preparation, characterization, and biochemical activities of N-(2-Carboxyethyl) chitosan from squid pens. Journal of agricultural and food chemistry, 63(9), 2464-2471. Nothing related to nanoparticles and tartaric acid in this reference.

Introduction Section needs to be re-written. The information presented is not obviously related to the detailed manuscript contribution to the knowledge in the field. The last paragraph, L88-L95, presents the conclusion of the manuscript and not its aim and scope.

It is mentioned, L125-L126, that ”The particle sizes were measured using Master Sizer 2000 laser granulometer”. However, I found only a mention in the Conclusion Section ”CS-TA-Na nanocrystals were 20-50 nm in length and 10-15 nm in width in solution” without presenting the supporting results.

Specific comments

L118. Resolution and average scan for the FTIR must be presented.

L204 SEM images. The length of the bar must be presented.

L285-L286. Reference 13 must be corrected. The entire References Section must be corrected.

Round 2

Reviewer 2 Report

The authors made some improvements. However, in my opinion, these improvements are not enough for manuscript  publication in the present form

The source of chitosan is still not mentioned. The FTIR measuring geometry is not mentioned - Transmission, Reflection/ATR?

Author opinion

Polymers journal indications

Regarding the writing of the introduction, our opinions differ significantly from those of the reviewer. The Introduction section should mainly introduce the knowledge in the field, but it should also appropriately introduce the work itself. No one stipulates that the introduction section must introduce the aim and scope of the paper. In fact, we only used a small amount of space to give a brief introduction to the whole work, the purpose is to attract readers to read this article in depth. The question raised by reviewer is just the way she/he think it should, and it does not mean that everyone must be consistent with her/him

The introduction should briefly place the study in a broad context and highlight why it is important. It should define the purpose of the work and its significance. The current state of the research field should be carefully reviewed and key publications cited. Please highlight controversial and diverging hypotheses when necessary. Finally, briefly mention the main aim of the work…

Therefore, I keep my recommendation – the last paragraph from the Introduction Section must be re-written and include the main aim of the work. Not because  I “think it should be” - because these are the indications of the journal wherein the authors choose to publish their paper.

The FTIR resolution indicates the minimum peak interval that can be distinguished. The resolution is set up before starting the work with the equipment. For a typical FTIR, the resolution is 4 cm−1. In the case of the equipment that you used, TENSOR 27 FT-IR spectrometer, Bruker,  the maximum resolution is “Better than 1 cm-1 (apodized), optional better than 0.5 cm-1  (apodized)”. Did you know the resolution that you set up for recording your spectra?  The average (number of) scans is the number of recording the same signal repeatedly - that allows averaging several acquisitions to reduce the noise by recording the same signal. Your spectra are smooth. Therefore the average scan was probably high – however, you did not mention it in the manuscript – and you asked me to “clearly indicate what “Resolution and average scan the FTIR” refer to.”

A superficial approach still weakens the manuscript. A scientific manuscript must keep its rigorous precision. In Figures 1, 2, 3, there is no Y-axis title. For example, most probably in Figure 1, the title of the Y-axis is Transmittance (%). In Figure 8, the bar length must be presented.

Round 3

Reviewer 2 Report

Manuscript was improved according to requirements.